# Arrhythmic Burden in Cardiac Amyloidosis: What We Know and What We Do Not

**DOI:** 10.3390/biomedicines10112888

**Published:** 2022-11-10

**Authors:** Alessia Argirò, Annamaria Del Franco, Carlotta Mazzoni, Marco Allinovi, Alessia Tomberli, Roberto Tarquini, Carlo Di Mario, Federico Perfetto, Francesco Cappelli, Mattia Zampieri

**Affiliations:** 1Tuscan Regional Amyloidosis Centre, Careggi University Hospital, 50134 Florence, Italy; 2Department of Internal Medicine I, San Giuseppe Hospital, 50053 Empoli, Italy; 3Structural Interventional Cardiology Department, Careggi University Hospital, 50134 Florence, Italy

**Keywords:** arrhythmias, atrial fibrillation, pacemaker implantation, cardiac amyloidosis

## Abstract

Cardiac amyloidosis (CA), caused by the deposition of insoluble amyloid fibrils, impairs different cardiac structures, altering not only left ventricle (LV) systo-diastolic function but also atrial function and the conduction system. The consequences of the involvement of the cardiac electrical system deserve more attention, as well as the study of the underlying molecular mechanisms. This is an issue of considerable interest, given the conflicting data on the effectiveness of conventional antiarrhythmic strategies. Therefore, this review aims at summarizing the arrhythmic burden related to CA and the available evidence on antiarrhythmic treatment in this population.

## 1. Introduction

Cardiac amyloidosis (CA) is an infiltrative disease characterized by the extracellular deposition of insoluble amyloid fibrils in the heart that lead to increased left ventricular (LV) wall thickness, impaired LV relaxation and a reduction in LV systolic function. Recent studies have clearly shown that CA, particularly transthyretin-related amyloidosis (ATTR), is a leading cause of heart failure (HF), affecting approximately 15% of subjects with HF with preserved ejection fraction (HFpEF) [1]. While several studies have clarified the morphological and functional consequences of amyloid deposition on cardiac structures [2,3], the impact of amyloid infiltration on the electrical conduction system of the heart and the arrhythmic profile of patients with CA has often been overlooked.

In CA patients, various arrhythmias can be detected (Figure 1) that are caused by several mechanisms, including inflammatory cell damage, cellular degradation and the separation of myocytes by amyloid fibrils [4]. Specifically, arrhythmias are the result of a combination of amyloid accumulation and the involvement of closed structures that might influence the cardiac complex balance. According to the different altered proteins, the type and the prevalence of a single arrhythmia can change. This is the case for atrial fibrillation (AF), which appears to be more frequent in wild-type ATTR (wtATTR) [5], while ventricular arrhythmias (VAs) have been mostly described in association with light-chain amyloidosis (AL) [6].

The presence of arrhythmias in CA patients is associated with a poorer prognosis, reflecting a higher risk of HF progression and mortality [7].

The aim of our review is to summarize the state of the art on the arrhythmic burden in CA and, most importantly, to highlight the controversies related to antiarrhythmic treatment in this population.

## 2. Atrial Arrhythmias

Atrial function is heavily influenced by the direct toxic effect of amyloid accumulation, even in the early stages of the disease [8]. Histological findings of extreme amyloid infiltration in the atria support the “adverse remodelling hypothesis” determining the loss of atrial architecture, the remodeling of the vessels, capillary disruption and an upregulation of collagen synthesis at the level of the atria [9].

Amyloid infiltration is associated with the deterioration of the three atrial phases: reservoir, conduit and contraction. Specifically, in CA, the atrial chamber behaves as a non-compliant reservoir during ventricular systole and acts as a poorly efficient contractile chamber during late ventricular diastole.

### 2.1. Prevalence

Atrial involvement in AL and ATTR is associated with a high burden of cardiac arrhythmias. The prevalence of AF in patients with CA is variable among studies, with the most recent reports assessing a very high prevalence, reaching almost two-thirds of the population [6,7]. Some differences among studies might be attributed to the presence of implanted cardiac devices determining a higher success in detecting subclinical AF [10]. However, nearly half of patients with AL or ATTR show concomitant AF at the time of diagnosis [11], and during follow-up, wtATTR was revealed to be a stronger predictor of AF over hereditary ATTR (hATTR), likely due to the lower age and earlier disease detection and treatment in hATTR [12].

### 2.2. Pathogenesis

Because of the increasingly high prevalence of AF in CA patients, it is reasonable to assume that AF and CA have a causal relationship. There are several mechanisms for the development of AF in CA. Firstly, amyloid deposition within atrial tissue electro-anatomically disrupts homogeneous electrical conduction, causing large areas of voltage attenuation [13]. Secondly, the direct toxic effect of amyloid fibrils on cardiomyocytes results in fibrosis and oxidative stress, which are powerful substrates for AF [14,15]. Thirdly, small vessel disease due to perivascular amyloid infiltration represents a likely substrate for myocardial ischemia [16]. Finally, it seems that AF itself contributes to progressive amyloid deposition, promoting LA myopathy [17].

### 2.3. Arrhythmia Detection

A proposed scheme for the follow-up of patients with CA suggests performing a yearly 24 h Holter electrocardiogram (ECG) [18]; this is based mainly on clinical practice, and an optimal follow-up scheme has yet to be defined. Almost half of patients manifest AF before the diagnosis of CA [11], while others might have later thromboembolic events without showing any previous symptoms [19]. In this context, patient monitoring with a prolonged Holter ECG or implantable loop recorders (ILRs) could be useful in the early identification of AF and, subsequently, in the prescription of anticoagulation.

### 2.4. Prognostic Implications

Whether or not AF impacts overall or cardiovascular mortality in CA is still unsolved [5,10,17]. The maintenance of sinus rhythm, obtained by either cardioversion or ablation performed in the early stages of the disease, seems to be more effective in improving symptoms and reducing hospitalization among CA patients [17]. However, the presence of normal sinus rhythm does not guarantee preserved atrial function. In this context, the detection of myocardial deformation by speckle tracking at echocardiography allows the identification of one-fifth of patients who show a severe impairment of contractility despite remaining in sinus rhythm on the electrocardiogram [20]. This condition is also known as atrial electromechanical dissociation, proven to be associated with a poor prognosis [9,21].

Moreover, speckle tracking allows the detection of differences not only in overt disease manifestation but also in the subclinical setting. Specifically, if, on one side, left atrial dysfunction proves to be more pronounced in ATTR (compared to AL) patients, despite left atrial volumes being comparable [22], on the other side, the detection of left atrial abnormalities in carriers with a transthyretin valine-to-isoleucine substitution underlines subtle left ventricular remodeling [23].

Evaluating atrial function before the possible detection of atrial arrhythmias might also be helpful in detecting an increased risk of thromboembolic events, which may happen even in sinus rhythm [24]. All of this evidence underlines the distinctive nature of AF in CA and the importance of a comprehensive and multiparametric evaluation of atrial function.

### 2.5. Stroke Risk and Anticoagulation

Patients with CA have been found to have an increased risk of developing intra-cardiac thrombi, even in sinus rhythm, likely due to atrial mechanical dysfunction, endothelial dysfunction and relative hypercoagulability [8,25].

Feng et al., in a large study carried out at the Mayo Clinic involving 116 autopsies of patients with CA (AL, ATTR and serum amyloid A), observed a prevalence of intracardiac thrombi of 33%, compared to none in the control group. The combination of AL and AF was associated with the highest risk of thrombus detection [26].

In addition, in a cohort of 156 patients with CA who underwent transesophageal echocardiography (TEE), intracardiac thrombi were detected in 27% of patients. In this population, AL patients more frequently showed intracardiac thrombi compared to ATTR patients (35% vs. 18%; *p* = 0.02) despite being younger and having less AF [27].

The prevalence of intracardiac thrombi assessed by cardiac magnetic resonance (CMR) was 6.2% in a study by Martinez-Naharro et al. [28] including 324 amyloidosis patients, both ATTR and AL. Favoring factors were biventricular systolic dysfunction, atrial dilation, AF, higher extracellular volume and AL subtype. Several other studies have demonstrated an increased incidence of intracardiac thrombi in patients with CA despite the absence of AF/flutter [25,29].

The incidence of arterial thromboembolic events in CA was described by Cappelli et al. [24] in a cohort of 406 patients, both AL and ATTR. Thirty-one patients (7.6%) suffered from thromboembolism, mainly cerebrovascular, of whom ten (32.2%) were in sinus rhythm and had no history of AF. In a larger, international, multicentric study, Vilches et al. [30] confirmed the high prevalence of embolic events in patients with ATTR, either with or without AF. In their cohort, CHA2DS2-VASc did not predict embolic events, suggesting its limited role in estimating the risk of thromboembolism in CA. Additionally, they did not find meaningful differences in the rate of embolism between patients with AF treated with a vitamin K antagonist (VKA) and those treated with novel oral anticoagulants (NOACs) [30].

There are limited data on the optimal anticoagulant strategy; specifically, little is known about the safety of NOACs in this population and whether differences exist in the occurrence of embolic events. Considering the advanced age of the subjects, both bleeding and thrombotic risks are generally perceived as high. Moreover, the VKA response is limited by inter/intra-patient variability and compliance with a complex medical regimen and diet, making treatment difficult in this population.

In a recent study, Mitrani et al. [31] found no difference in the combined outcome of stroke, transient ischemic attacks (TIA), major bleeding or death in patients with ATTR and AF treated with either VKA or NOACs. Maintaining the international normalized ratio (INR) in the normal range appears to be crucial, since all patients on VKA with a stroke or TIA showed a labile INR. Additionally, the higher bleeding risk was confined to the same subset of patients with a labile INR.

Cariou et al. [32] compared 147 (54%) vs. 126 (46%) patients receiving VKA and NOACs, respectively. In the wtATTR subgroup, patients receiving VKA had a higher bleeding risk compared to patients on NOACs (major bleeding events in 14 vs. 2%, respectively; *p* < 0.001), but there was no significant difference in ischemic events. In the AL subgroup, the bleeding risk was similar between groups, and not a single stroke was registered.

In contrast to the study by Mitrani et al. [31], Cariou et al.’s cohort [32] showed a higher bleeding risk in patients on VKA, which may have been driven by their more impaired renal function; however, both these retrospective studies showed that NOACs can be used safely in CA.

In conclusion, there is a high prevalence of atrial thrombosis in both AL and ATTR. CA has been shown to expose patients to an increased risk of embolic events, and this risk is not limited to patients with clinical AF [25]. At present, there are no strong data to recommend which oral anticoagulant to prefer, although NOACs have proven to be safe.

### 2.6. Our Point of View

Patients with CA show a high propensity to develop intracardiac thrombi and embolic events, even without evidence of AF. However, no clear guidelines have been provided so far that will help the clinician in the management of anticoagulation therapy in patients with CA and no AF. Patients with CA are often old and frail, predisposing them to an increased risk of bleeding; however, an embolic event may dramatically reduce their clinical and performance status and their quality of life. In many situations, clinicians will face decisions on anticoagulation therapy, and thus, we would like to provide a little insight into our clinical practice that may provide some suggestions to other physicians (Table 1).

Firstly, routine Holter monitoring is fundamental to screen patients for concealed episodes of AF. As per standard practice, we perform a Holter ECG on a 6-month basis to detect asymptomatic AF, allowing us to introduce anticoagulation irrespectively of the CHA_2_DS_2_-VASc score. In addition, at diagnosis, we usually perform cardiac magnetic resonance, which has been revealed to be a useful tool in both providing tissue characterization and identifying possible intracardiac thrombi.

During follow-up evaluation, it is not infrequent to identify patients presenting severe atrial enlargement or “atrial standstill” or the presence of spontaneous echo contrast within the atria at echocardiography. Atrial standstill might be defined by the absence of mechanical activity in the atria, as assessed visually at echocardiography, or by using atrial strain, and it might be associated with the presence of a low mitral inflow A-wave amplitude. In a few patients, these features are associated with the presence of a clear P wave on the electrocardiogram (in the absence of a clinical history of AF). These conditions raise many concerns about the risk of thrombus formation. In these cases, we actively try to identify evidence for anticoagulation treatment, i.e., reducing the interval between Holter ECG evaluations, implanting a loop recorder or providing a home monitoring device for patients with a pacemaker or ICD, and, when renal function allows us, we repeat cardiac magnetic resonance with the aim of atrial thrombus identification.

In conclusion, we think that in the absence of clear guidelines, it is still controversial to initiate anticoagulation therapy without evidence of AF or without identifying a thrombus. However, in the presence of echocardiographic signs of an increased risk of thromboembolism, a more aggressive and proactive approach in order to identify asymptomatic AF or signs of atrial thrombosis could be reasonable.

### 2.7. Rate Control

Rate control is particularly challenging in CA, mostly due to the coexistence of autonomic dysfunction and restrictive cardiac physiology with a low and relatively fixed stroke volume. In this scenario, a higher heart rate is often necessary to maintain an adequate cardiac output [33].

Non-dihydropyridine calcium channel blockers are contraindicated in CA for their negative inotropic/chronotropic effect and the high risk of hypotension [14]. Beta-blockers may also be poorly tolerated; however, low doses of beta-blockers may be an option to achieve rate control in AF with a rapid ventricular response [34].

The role of digoxin in CA remains controversial. Historically, Rubinow et al. [35] showed that digoxin binds avidly to amyloid fibrils in vitro, suggesting a higher risk of digoxin toxicity. A more recent study re-evaluated digoxin’s utility in the rate control strategy in 69 patients with CA. Although suspected digoxin-related arrhythmias and toxic events occurred in 12% of patients, no deaths were attributed to digoxin toxicity [36]. Thus, low-dose digoxin with close monitoring is a possible alternative for rate control in selected patients, especially when other therapeutic strategies are limited by hypotension.

Finally, in the case of failure to obtain rate control with medical treatment, atrioventricular nodal ablation and a permanent pacemaker implant may be considered [37].

### 2.8. Rhythm Control

The loss of the atrial contribution to ventricular filling in AF often leads to the patient’s clinical deterioration. Rhythm control by means of direct current cardioversion (DCCV) has been recently described with variable success and recurrence rates [5,11].

In the study by El-Am et al. [43], patients with scheduled DCCV and CA suffered from a significantly higher DCCV cancellation rate compared to patients with AF without CA, mostly due to the identification of intracardiac thrombi. Thus, TEE should be performed before DCCV in all patients with CA, regardless of the duration of AF or anticoagulation status [25,26,27,43].

The rate of success of DCCV was high (90%) and similar between patients with and without CA. Furthermore, the incidence of arrhythmia recurrence during a 1-year follow-up was also high but similar between the two groups (48% vs. 55%; *p* = 0.75). However, the procedural complication rate was significantly higher in the CA group (14% vs. 2%, respectively, *p* = 0.007), reflecting the underlying advanced myopathic and electrical disturbances in CA [43].

Similarly, Donnellan et al. [17] reported a retrospective analysis on 256 patients with ATTR and AF: 119 (45%) patients underwent DCCV, and sinus rhythm was initially restored in 113 (95%) of them and appeared more effective when performed earlier in the disease course. One year after DCCV, 49 (42%) patients remained in sinus rhythm, and, interestingly, the maintenance of sinus rhythm was significantly associated with lower mortality (43% vs. 69%, *p* = 0.003).

In summary, although DCCV is very effective in restoring sinus rhythm, the recurrence rate of atrial arrhythmias is high. However, DCCV appears to be an appealing approach in the early stages of the disease.

Given the limitations of rate control strategies, a rhythm control strategy may be considered for the management of AF, particularly for earlier disease stages. In a retrospective analysis of wtATTR, Mints et al. described 33 patients who received antiarrhythmic treatment, mainly amiodarone, for AF and found no survival benefit from rhythm control compared to the rate control strategy [5].

Little is known about the safety and efficacy of AF or flutter ablation in patients with ATTR.

Only a few small retrospective studies have examined the role of catheter ablation of atrial arrhythmias in patients with CA, with inconsistent results [12,44,45].

Tan et al. [44] reported results on a retrospective cohort including 13 patients, both AL and ATTR, who underwent atrial arrhythmia ablation, of whom 5 had AF; the 3-year recurrence-free rate was 60% for all atrial arrhythmias and 40% for AF, and ablation was associated with a reduction in the New York Heart Association functional class (NYHA) in 70% of cases.

More recently, Donnellan et al. [45] reported the largest cohort of radiofrequency ablation in 24 patients with ATTR. During a mean follow-up of 39 months, the overall recurrence rate of AF was 58%, and among patients who developed recurrent arrhythmias, the AF-free mean time from ablation was 23 months. Ablation appeared less effective in those with a higher ATTR stage, older age and higher NYHA class. However, the rate of hospitalization for AF or HF was markedly lower in patients who underwent atrial arrhythmia ablation, and after a follow-up of more than 3 years, ablation was associated with improved survival.

After catheter ablation, the long-term maintenance of sinus rhythm appears to be frequently difficult, especially in more advanced stages of the disease. However, in a selected group of patients with an early stage of ATTR, AF ablation might be reasonable to reduce recurrent hospitalization for symptomatic AF or HF.

Moreover, it appears reasonable to associate CA target therapy with catheter ablation. From a sub-analysis of data from the Transthyretin Amyloidosis Cardiomyopathy Clinical Trial (ATTR-ACT), tafamidis—a medication used to delay disease progression in adults with ATTR-CA—contributed to reducing hospitalization due to arrhythmias [46]. Specifically, the percentage of patients in SR was higher in those who took tafamidis after ablation therapy than those who did not [16].

All of the above studies encountered many limitations, such as the small number of patients and the few data comparing patients with CA and those without. Therefore, currently, we do not have therapeutic approaches designed selectively for the patient with CA.

### 2.9. Atrial Fibrillation in Heart Failure with Preserved Ejection Fraction and Cardiac Amyloidosis

Atrial fibrillation and HFpEF are strictly related and considered “vicious twins” [47]. Indeed, patients with AF present a 4.8 times higher risk of developing HFpEF compared to patients in sinus rhythm [48], and the prevalence of AF in HFpEF is high, ranging between 15% and 41% [47]. Furthermore, as we have seen in wtATTR, two-thirds of patients with solely HFpEF experience AF over time [49]. As in cardiac amyloidosis, AF and HFpEF are manifestations of a common atrial and ventricular myopathy. While in CA, the deposition of amyloid fibrils plays a pivotal role, in patients with solely HFpEF, systemic inflammation and metabolic disorders may lead to microvascular dysfunction and fibrosis of both atria and ventricles, which in turn trigger diastolic dysfunction and AF [47]. Furthermore, HFpEF and AF feed off each other. As happens in CA, the presence of diastolic dysfunction and elevated LV filling pressure contributes to LA enlargement and electrical remodeling and eventually leads to AF [50]. Moreover, the presence of AF in both solely HFpEF and CA may worsen HF symptoms, probably due to the loss of LV filling mediated by the atrial kick [51,52].

## 3. Atrioventricular Conduction Diseases

Conduction system diseases are frequent in CA, and the prevalence of pacemaker implantation at diagnosis ranges between 8.9% and 10% [53,54].

From an electrophysiological standpoint, patients with CA present a prolonged Hiss-ventricle (HV) conduction interval compared to patients without CA [12], and the HV conduction delay is more profound in ATTR than in AL.

On a surface electrocardiogram at diagnosis, first-degree atrioventricular (AV) block was more frequent in wtATTR compared to AL, while the presence of an intraventricular conduction delay was more common in both wtATTR and hATTR compared to AL [54]. The frequency of right bundle branch block is similar between ATTR and AL, whereas left bundle branch block seems to be uncommon in AL [7]. Indeed, the long slender right bundle branch may be more vulnerable to amyloid deposition compared to the left, and therefore, even a low amount of amyloid deposition, characteristic of AL-CA, may impact its electric impulse conduction [55,56].

### 3.1. Pacemaker and Loop Recorder

Fifteen percent of patients with hATTR and 30% of patients with wtATTR already had a pacemaker implanted at diagnosis compared to only 1% of patients with AL [54]. However, in a multicenter retrospective study including 405 patients, during a median follow-up of 33 months, the incidence of pacemaker implantation was similar among amyloidosis subtypes (8.9% during a median follow-up of 33 months), raising the suspicion that the pathophysiology underlying conduction disturbances may be different between AL and ATTR.

In ATTR, the main mechanism leading to conduction abnormalities seems to be progressive amyloid deposition that alters the myocardial structure and undermines electrical conduction [57]. In a CMR study from the National Amyloidosis Centre in London, patients with ATTR presented greater LV mass and amyloid deposits, as expressed by the extracellular volume, compared to AL [56]. Yet, patients with AL showed higher native T1 mapping compared to ATTR due to a greater amount of myocardial edema [56]. These findings were confirmed by a following study in which patients with untreated AL showed the greatest increase in myocardial T2 (a CMR biomarker of myocardial edema) compared to treated AL and ATTR [58]. Therefore, the cytotoxicity of free light chains [59] may lead to conduction disturbances, following the model of myocarditis in which edema plays an important role in arrhythmogenicity [60].

Independent predictors of PM implantation include a history of AF, PR interval > 200 ms and QRS > 120 ms. The highest risk of PM implantation emerged with the coexistence of all three parameters in both AL and ATTR (hazard ratio 6.26, CI 1.9–20.6). ATTR patients presenting these electrocardiographic predisposing factors showed signs of more advanced disease, such as a greater LV thickness and worse biventricular systolic function, whereas no differences emerged in the distribution of the Mayo score for AL patients [53].

Data from the longitudinal pacemaker interrogation in patients with CA showed a progressive increase in the mean ventricular pacing, and, while the pacing burden was 56% at 1 year post-implantation, most patients at 5 years showed near 100% ventricular pacing [61]. Furthermore, over time, the right ventricular sensing amplitudes decreased, but lead impedances and capture thresholds were stable in the absence of device malfunction [61].

The role of internal loop recorders (ILRs) for the early detection of bradyarrhythmias has still to be clearly defined. Sayed et al. implanted ILRs in 20 consecutive patients with symptoms of syncope or presyncope and advanced AL. Interestingly, death was preceded by bradycardia, complete atrioventricular block and the development of pulseless electrical activity (PEA). A pacemaker was implanted in four patients due to AV block, yet three of them, who were previously resuscitated from PEA, died anyway [62]. They hypothesized that severe bradycardia and AV block may further reduce an already impaired cardiac output, resulting in ischemic damage that may lead to further decompensation and PEA. Presumably, a narrow time window for intervention exists; indeed, the only patient who received a pacemaker before a significant reduction in cardiac output survived.

Interestingly, in a recent study, the role of a prophylactic pacemaker was tested in patients with hATTR and slowed AV conduction, and the pacemaker prevented major cardiac events in 25% of them during a follow-up of 45 months [63]. However, high-grade AV block was not independently associated with mortality after adjusting for the disease stage and the presence of coronary artery disease [64].

In conclusion, patients with CA often require pacemaker implantation due to progressive amyloid deposition that alters the electrical conduction system, and advanced conduction system disease may represent a relevant competing cause of death in CA.

### 3.2. Resynchronization Therapy

In ATTR, right ventricular pacing > 40% has been shown to be associated with worsening mitral regurgitation, reduced LV ejection fraction and worsening HF symptoms compared to patients with biventricular pacing [65]. Furthermore, cardiac resynchronization therapy has been associated with reduced all-cause mortality and cardiovascular hospitalization.

However, more data are needed to confirm these preliminary findings and to define the subgroup of patients that benefit more from this treatment [18].

## 4. Ventricular Arrhythmias

Data regarding VAs in CA are scarce if compared to the other electrophysiological manifestations of the disease and are mainly derived from small retrospective studies.

### 4.1. Prevalence

In ATTR, the estimated prevalence of non-sustained ventricular tachycardias (NSVT) is between 17% and 20% on Holter monitoring [66,67], while in AL-CA, the prevalence ranges from 5 to 29% [34,66,68]. However, in AL-CA, NSVTs might be more frequent, especially during the stem-cell transplantation period, as demonstrated by a small study conducted on 24 patients with telemetry monitoring during autologous stem-cell transplantation: NSVT was recorded in all patients and was the most common arrhythmia, and one patient experienced sustained VAs that required direct current defibrillation [69].

The prevalence of NSVT, in both ATTR and AL, increases up to 74% when analyzing long-term monitoring devices, such as a pacemaker or ICD, while approximately 20% of patients with a pacemaker or ICD experienced sustained VAs [70].

### 4.2. Pathogenesis

Patchy amyloid fibril deposition in the myocardium leading to an inflammatory response and oxidative stress results in a separation of myocytes, resulting in LV fibrosis, which progressively develops arrhythmogenic potential. In combination with this, amyloid fibril deposition at the conduction system level could potentiate arrhythmias, favoring the development of re-entrant circuits [71,72]. Additionally, microvascular ischemia (due to amyloid perivascular infiltration) and the direct cytotoxic effect of amyloid fibrils are held responsible for the genesis of VAs in CA [71,72]. The potential synergistic effect of the AL toxic effect along with drug-induced cardiac toxicity occurring during chemotherapy could further contribute to the genesis of VAs in patients with AL [6].

Myocardial amyloid infiltration can be easily identified with MRI imaging with increased T1 mapping, ECV and areas of LGE (subendocardial or transmural). All of these parameters have been demonstrated to have both diagnostic and prognostic implications, with transmural LGE and higher ECV linked to a greater risk of all-cause mortality. Unlike other forms of cardiomyopathy, though, evidence of a correlation between LGE or ECV and arrhythmic risk in CA is lacking [73,74,75].

### 4.3. Prognostic Implications

Although VAs are common in CA, their effect on cardiovascular mortality is still a matter of debate. Patients with CA mostly die of worsening HF, and the mechanism of sudden cardiac death has traditionally been attributed to electromechanical dissociation rather than VAs [38,62,71], questioning the benefits of ICDs.

The prognostic role of NSVT in CA is controversial. Some studies suggested their association with sudden cardiac death [68,70], while others hypothesized that NSVT may represent a marker of disease severity rather than a predictor of sudden arrhythmic death [62,66].

Nonetheless, recent evidence suggests that the impact of VAs on CA patients may have been undervalued, and they may represent a frequent competing cause of death in both AL and ATTR.

In a cohort of 5585 patients hospitalized for CA, 2020 (36%) had concurrent arrhythmias, and ventricular tachycardia was the second most common arrhythmia identified (14.9%), after AF (72.2%). All-cause mortality and HF were significantly higher in patients with CA hospitalized with concurrent arrhythmias compared to those without [76].

Regarding the specific subset of AL, in a retrospective cohort of 56 patients, 8 experienced sudden cardiac death (interestingly, almost all episodes occurred during chemotherapy), with VAs being the presenting rhythm in 4 cases; PEA was observed in just 1 patient, while the presenting rhythm of the remaining 3 patients was unknown [77].

### 4.4. Sudden Cardiac Death, Pharmacological Treatment and ICD

The pharmacological management of VAs in CA is essentially limited to amiodarone and, if tolerated, to small doses of beta-blockers [40].

Non-pharmacological therapy is mainly based on ICDs; however, the indications and timing for a primary prevention ICD are still a matter of debate.

Patients with CA (both AL and ATTR) tend to have a worse prognosis than other forms of HF, and traditional thresholds for a primary prevention ICD, such as ejection fraction <35%, are scarcely adequate in the context of CA, where systolic dysfunction is a hallmark of very advanced disease with often limited life expectancy [41,75].

Furthermore, the most recent European Guidelines on the management of ventricular arrhythmias refer specifically to CA only for ICD implantation in patients with hemodynamically non-tolerated VT and stress the importance of careful discussion with patients about other possible causes of cardiac and non-cardiac death [78].

A registry study of 472 patients with CA and an ICD found a mortality rate of 26.9% at 1 year after ICD implantation compared with 11.3% among a propensity-matched cohort of patients with other non-ischemic cardiomyopathies, and CA was also associated with a significantly higher risk of all-cause mortality. A history of syncope, VAs, diabetes mellitus and cerebrovascular disease were factors associated with a higher risk of death within 1 year from ICD implantation [42].

A case–control study comparing 23 patients with CA and a primary prevention ICD to patients with CA without an ICD and patients with a primary prevention ICD for ischemic or non-ischemic cardiomyopathies showed comparable rates of appropriate ICD therapies between amyloid and non-amyloid patients. However, the presence of an ICD was not associated with longer survival when compared to CA patients without an ICD. Furthermore, patients with CA and an ICD had a significantly higher mortality rate than the non-amyloid ICD recipients [39].

Similarly, in a cohort of 130 patients with mainly hATTR (67%) and a high rate of systolic HF (62%), the incidence of VAs was high (53%, mostly NSVT). In the 32 patients with an ICD implanted for primary prevention, the rate of appropriate ICD therapy was 25%. However, no significant survival benefit was found upon comparison with similar ATTR groups without ICDs [79].

In conclusion, an ICD does not seem to have a pivotal role in extending life expectancy in CA; however, new and effective therapies are becoming progressively available for the treatment of both AL and ATTR. Life expectancy in CA patients will hopefully increase in the near future, likely making ICDs more impactful on survival. For now, careful patient selection and shared decision making are of the outmost importance when deciding on ICD implantation in a patient with CA.

## 5. Conclusions

The presented data underline a unique phenotype of cardiac remodeling associated with CA, with the progressive involvement of conduction tissue and corresponding arrhythmic expression. With the advancement of the CA stage, AF, conduction disorders and ventricular arrhythmias become more pronounced and are associated with worse survival. In this regard, it is important to detect any predictable electrical disorders early and also define a treatment based on comorbidities and symptoms. Whereas clinicians rely on device therapy for bradyarrhythmia, unfortunately, the optimal treatment strategy for AF, stroke or the ventricular arrhythmic burden remains an issue of high clinical relevance that needs to be addressed. Amyloid-specific and disease-modifying therapies could potentially play a key role in this context, possibly changing the electrical phenotype associated with CA and improving outcomes.

## Figures and Tables

**Figure 1 biomedicines-10-02888-f001:**
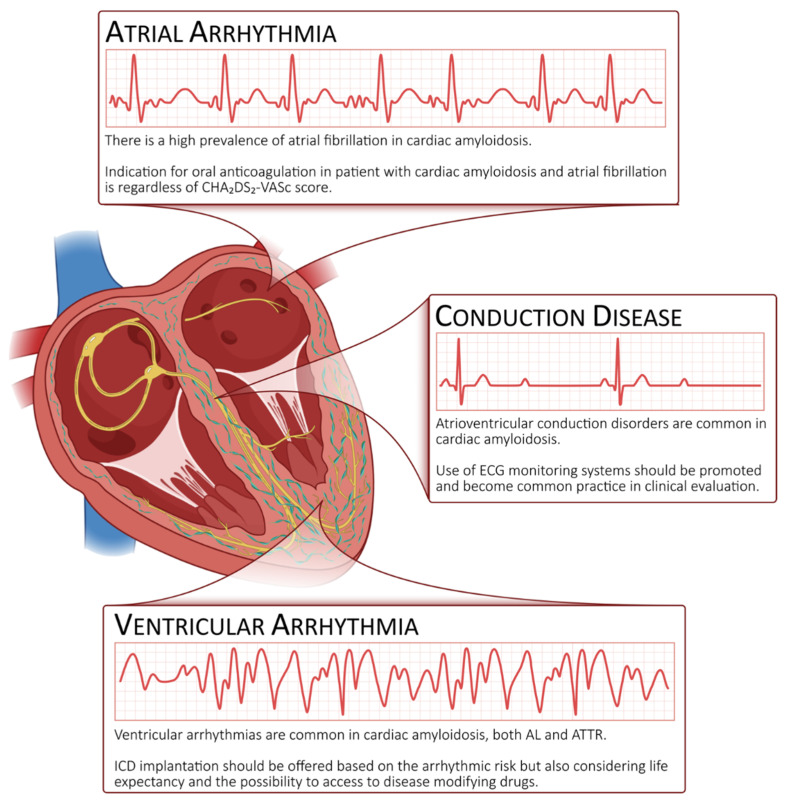
The electrical phenotype of cardiac amyloidosis is characterized by 3 possible scenarios: atrial fibrillation, conduction disease and ventricular arrhythmias.

**Table 1 biomedicines-10-02888-t001:** Pros and Cons of standard management of atrial fibrillation and ventricular arrhythmias in CA patients.

Treatment	Pros	Cons
Anticoagulation	- Increased risk of developing intracardiac thrombi [8,25] - Increased incidence of intracardiac thrombi despite the absence of AF/flutter [22,23,24,24,25,26] - Intracardiac thrombi more frequently in AL, biventricular systolic dysfunction, atrial dilation, AF and higher extracellular volume [24]	- CHA2DS2-VASc showed limited role in estimating the risk of thromboembolism in CA [27] - High frailty
VKA vs. NOACs	- No differences in the rate of embolism between VKA and NOACs [27,28]	- VKA had a higher bleeding risk [29]
Rate control	- Low doses of beta-blockers in AF with rapid ventricular response [31] - Low-dose digoxin with close monitoring is a possible alternative [33] - AVN ablation and PM implant may be considered when rate control is not obtained with pharmacological therapy [34]	- Coexistence of autonomic dysfunction and low stroke volume [30] - Non-dihydropyridine calcium channel blockers are contraindicated [10]
Current cardioversion	- High rate of success [35] - More effective when performed in the early stages of disease [12] - Maintenance of SR was associated with lower mortality [12]	- Frequent intracardiac thrombi [35] - High rate of procedural complications [35] - High recurrence rate [35]
Rhythm control	- NYHA reduction in 70% of cases [36] - Lower hospitalization rate following AF ablation [37] - AF ablation was associated with improved survival [37]	- No survival benefit from rhythm control (amiodarone) compared to rate control strategy [35] - Less effective in higher ATTR-CA stage, older age and higher NYHA [37] - High recurrence rate [37]
ICD implantation	- VAs are common in CA patients, and the rate of appropriate ICD therapies is similar to that of other forms of HF [38,39] - New specific therapies for amyloidosis may prolong life expectancy, making ICD impactful on survival	- Traditional thresholds for ICD implantation in primary prevention are scarcely applicable in CA [40,41] - No significant survival benefit demonstrated [39,42]

## Data Availability

Not applicable.

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
