# Peer review of "Arrhythmic Burden in Cardiac Amyloidosis: What We Know and What We Do Not"

_biomedicines, 2022, doi:10.3390/biomedicines10112888_

Round 1
Reviewer 1 Report (Previous Reviewer 3)
I have no further comments
Author Response
Thank you for your revision. We appreciate your interest in our work
Reviewer 2 Report (New Reviewer)
This review summarizes state-of-the-art on the arrhythmic burden in cardiac amyloidosis (CA), including atrial arrhythmias, atrioventricular conduction diseases, and ventricular arrhythmias. Meanwhile, they highlight the controversies related to the treatment and put forward some of their perspectives.
This review is interesting and valuable because it guides clinical diagnosis and treatment by summarizing the current research progress. However, a few changes must be made, as listed below.
Major points:
1. The abstract and the introduction are nearly identical. The authors should introduce CA and the association between CA and arrhythmia in detail.
2. The underlying mechanism of how CA contributes to arrhythmia pathogenesis should be discussed in detail.
3. Table 1. The treatments mentioned in the review seem non-specific between CA and non-CA patients. The difference in treatment between CA and non-CA patients should be highlighted in Table 1.
4. Though the title is “arrhythmic burden in cardiac amyloidosis”, most of the treatment strategies are targeting arrhythmia. Are there any treatment strategies against CA? Could these treatments help to alleviate arrhythmia? This should be included in the review.
5. On Page 2, line 87: Why the conclusion of left ventricular remodeling is drawn from the detection of left atrial abnormalities.
6. The conclusion is missing at the end of the article.
7. On Page 12: Why are there two paragraphs with the same description of the abbreviation?
8. On Page 4, line 167: Please specify why “In few patients these features are associated to the presence of a clear P wave on the electrocardiogram”. As far as I know, atrial standstill also has no electrical activity aside from the cessation of atrial mechanical activity.
9. On Page 4, line 179, and Page 5, line 221: Direct current cardioversion, drug cardioversion, and catheter ablation are all rhythm control techniques, thus it is unnecessary to discuss them separately in different sections. In addition, the authors only talked about amiodarone for drug cardioversion, are there any studies on the use of propafenone and other medications?
10. Internal loop recorders are used to monitor the disease, while pacemakers and resynchronization therapy are used to treat it in cases of atrioventricular conduction disorder. Thus, pacemakers should be discussed with resynchronization therapy in one section instead of internal loop recorders.
Minor points:
1. On Page 8, lines 386, "both AL and TTR," the "TTR" is spelled incorrectly.
2. On Page 12, line 444, the complete name of "HFpEF" is misspelled in the abbreviated part.
3. Please standardize the placement of references in the text before or after punctuation, particularly at the end of paragraphs.
4. On Page 2, line 87, Please include the full name whenever the "TTR" first appears in the article.
5. On Page 10, lines 421, “Table 1” appears repeatedly.
6. There are a few typos in how the text uses spaces.
Author Response
Thank you for your revision, here are our point-by-point replies
Major points:
- The abstract and the introduction are nearly identical. The authors should introduce CA and the association between CA and arrhythmia in detail.
We applied you suggestions.
- The underlying mechanism of how CA contributes to arrhythmia pathogenesis should be discussed in detail.
We added some data on already included paragraphs.
- Table 1. The treatments mentioned in the review seem non-specific between CA and non-CA patients. The difference in treatment between CA and non-CA patients should be highlighted in Table 1.
We thank the referee for this suggestion; anyway at now there are not dedicated studies to manage arrhythmia or ICD implant in CA compared to non-CA patients. Therefore, indications are expert opinion based on and derived from HF patients.
- Though the title is “arrhythmic burden in cardiac amyloidosis”, most of the treatment strategies are targeting arrhythmia. Are there any treatment strategies against CA? Could these treatments help to alleviate arrhythmia? This should be included in the review.
There are preliminary data, not yet conclusive. We mentioned the study that suggests a potential role of tafamidis in promoting maintenance od sinus rhythm in ATTR-CA.
- On Page 2, line 87: Why the conclusion of left ventricular remodeling is drawn from the detection of left atrial abnormalities.
In that case the atrial abnormalities could imply an initial damage to the left ventricle (LV), thus suggesting to pay more attention also to the LV morpho-functional study.
- The conclusion is missing at the end of the article.
We included paragraph for conclusion.
- On Page 12: Why are there two paragraphs with the same description of the abbreviation?
We corrected this.
- On Page 4, line 167: Please specify why “In few patients these features are associated to the presence of a clear P wave on the electrocardiogram”. As far as I know, atrial standstill also has no electrical activity aside from the cessation of atrial mechanical activity.
This agrees with the know data regarding the non-simultaneous manifestation of the altered mechanical and electrical activity. Specifically, the mechanical atrial dysfunction might precede the absence of P wave at ECG.
- On Page 4, line 179, and Page 5, line 221: Direct current cardioversion, drug cardioversion, and catheter ablation are all rhythm control techniques, thus it is unnecessary to discuss them separately in different sections. In addition, the authors only talked about amiodarone for drug cardioversion, are there any studies on the use of propafenone and other medications?
Thank for you suggestions regarding the exposure method. Unfortunately, the aren’t current data on the use of other anti-arrhythmic drugs in this population.
- Internal loop recorders are used to monitor the disease, while pacemakers and resynchronization therapy are used to treat it in cases of atrioventricular conduction disorder. Thus, pacemakers should be discussed with resynchronization therapy in one section instead of internal loop recorders.
We chose this exposure association to highlight the usefulness of the ILRs in predicting who needs pacing therapy.
Minor points:
- On Page 8, lines 386, "both AL and TTR," the "TTR" is spelled incorrectly.
We corrected this.
- On Page 12, line 444, the complete name of "HFpEF" is misspelled in the abbreviated part.
According to the current ESC guidelines for heart failure (doi: 10.1093/eurheartj/ehab368), the abbreviation for heart failure with preserved ejection fraction is HFpEF.
- Please standardize the placement of references in the text before or after punctuation, particularly at the end of paragraphs.
We corrected this.
- On Page 2, line 87, Please include the full name whenever the "TTR" first appears in the article.
We corrected this.
- On Page 10, lines 421, “Table 1” appears repeatedly.
We corrected this.
- There are a few typos in how the text uses spaces.
Please confirm understanding: comment is about the use of double space instead of a single one, is it right?
Round 2
Reviewer 2 Report (New Reviewer)
Although the manuscript is importantly improved after revision, most of the issues the reviewer addressed were still unresolved. Therefore, the manuscript in the present form is not acceptable.
1. The authors should introduce CA and the association between CA and arrhythmia in detail in the introduction part.
2. I can’t find the revised sections regarding major point 2. Can you highlight the changes to your manuscript, and indicate which lines in the MS were changed in the point-by-point letter.
3. Regarding major point 3, the authors stated that there are no dedicated studies to manage arrhythmia or ICD implants in CA compared to non-CA patients. However, this point is very important, and should be discussed in the “Discussion” part based on current knowledge.
4. Regarding major point 5, the literature cited draws the conclusion that the abnormalities of left atrial (LA) shape and function rather than left ventricular remodeling are impacted by the transthyretin (TTR) valine-to-isoleucine substitution (V122I) mutation. In addition, why change “TTR” to “ATTR”, TTR in this literature represents “transthyretin”.
5. “HFpEF” is “Heart failure with preserved ejection fraction”, but not “Hear failure with preserved ejection fraction” in the “Abbreviations” part.
Author Response
- The authors should introduce CA and the association between CA and arrhythmia in detail in the introduction part.
We added a paragraph in the introduction regarding the association between CA and arrhythmias (lines 43-52)
- I can’t find the revised sections regarding major point 2. Can you highlight the changes to your manuscript, and indicate which lines in the MS were changed in the point-by-point letter.
We talked about the pathophysiology of arrhythmias in each section, we also made some changes regarding ventricular arrhythmias (lines 380-384).
- Regarding major point 3, the authors stated that there are no dedicated studies to manage arrhythmia or ICD implants in CA compared to non-CA patients. However, this point is very important, and should be discussed in the “Discussion” part based on current knowledge.
The table is only meant to summarize the main differences in the treatment of arrhythmias between CA and non-CA patients which are discussed in detail in every section of the manuscript. Unfortunately, evidence is often derived from very small studies: we tried to clarify this aspect for the management of atrial fibrillation by adding a sentence (lines 272-274). The controversies of ICD placement and the management of VAs in patients with CA vs non-CA patients are extensively discussed in paragraph 4.4 (lines 427-460).
- Regarding major point 5, the literature cited draws the conclusion that the abnormalities of left atrial (LA) shape and function rather than left ventricular remodeling are impacted by the transthyretin (TTR) valine-to-isoleucine substitution (V122I) mutation. In addition, why change “TTR” to “ATTR”, TTR in this literature represents “transthyretin”.
Thank you for highlighting the mistake in the concept explanation. The substitution is referred to transthyretin and we replied the abbreviation with it.
- “HFpEF” is “Heart failure with preserved ejection fraction”, but not “Hear failure with preserved ejection fraction” in the “Abbreviations” part.
Thank you for the input, we corrected the mistake in the “Abbreviations” part.
Round 3
Reviewer 2 Report (New Reviewer)
The reviewer thinks that the previous questions were answered quite well and the manuscript was revised carefully. This reviewer has no further comments.
This manuscript is a resubmission of an earlier submission. The following is a list of the peer review reports and author responses from that submission.
Round 1
Reviewer 1 Report
This review manuscript had chosen an interesting topic. It was well organized. The citation of references is appropriate and comprehensive. The main conclusion and comments are useful to the clinical and scientific society. This reviewer has the following comments or suggestions to prove the quality of the manuscript:
1) As the author mentioned in the introduction, CA-ATTR is affecting approximately 15% of HFpEF patients, it is highly suggestive to add the current knowledge in AF in HFpEF with or without CA.
2) The authors have talked about the controversies in the subtype specific response of CA to various treatments. It is highly suggested to use a table to summarize all these comparisons, which will benefit the readers to quickly get the information.
Author Response
....
Reviewer 2 Report
Excellent review article. Well written and organized. One thing the authors could do to enhance the article is provide diagrams or figures on some of the topics.
Author Response
.....
Reviewer 3 Report
This is a very relevant and wel-written review. I therefore have one some minor comments to make
1. atrial arrythmias: it would be nice to pay a little bit more attention to :
a: the use of LA echocardiography in relation to AF/AT diagnosis and their role on prognosis as there are substantial differences between AL and ATTR patients. please refer in this perspective to a very recent paper in biomedicines (pmid 35892668) and pmid 33928732 in mutation carriers.
b: anticoagulation in SR vs AF/AT. It would be interesting to receive some guidance from the authors i.e. a pro/con table. Do the authors consider atrial function / dysfunction like A wave on echo helpful to guide in the decision making?
ventricular arrythmias
a; please also refer to the novel ESC guideline on VTs that is just published. the section on cardiac amyloidosis has not changed. could the authors give some more guiding?
b. in this perspective please also describe or comment on the role of the ECV / scar tissue on MRI and risk stratication for VT; does it play a role in clinical decision making?
Author Response
.....